# Self-Supervised Speech Recognition via Local Prior Matching

## Abstract

We propose local prior matching (LPM), a self-supervised objective for speech recognition. The LPM objective leverages a strong language model to provide learning signal given unlabeled speech. Since LPM uses a language model, it can take advantage of vast quantities of both unpaired text and speech. The loss is theoretically well-motivated and simple to implement. More importantly, LPM is effective. Starting from a model trained on 100 hours of labeled speech, with an additional 360 hours of unlabeled data LPM reduces the WER by 26% and 31% relative on a clean and noisy test set, respectively. This bridges the gap by 54% and 73% WER on the two test sets relative to a fully supervised model on the same 360 hours with labels. By augmenting LPM with an additional 500 hours of noisy data, we further improve the WER on the noisy test set by 15% relative. Furthermore, we perform extensive ablative studies to show the importance of various configurations of our self-supervised approach.

## 1 Introduction

Fully supervised learning remains the mainstream paradigm for state-of-the-art automatic speech recognition (ASR). These systems require huge annotated data sets (Li et al., 2017; Chiu et al., 2018; Hannun et al., 2014; Amodei et al., 2016), which are time-consuming and expensive to collect. This hinders the development of accurate ASR systems for low resource languages (Precoda, 2013). In fact, out of over 6,000 spoken languages, fewer than 150 are supported by commercial ASR service providers. In sharp contrast to how we teach machines to recognize speech, humans do not learn by listening to thousand hours of speech and simultaneously reading the corresponding transcriptions. Instead, as noted in (Chomsky, 1986; Kuhl, 2004; Glass, 2012; Dupoux, 2018), humans possess an inherent ability to learn from vast quantities of unlabeled speech. Consider the case of conversing with someone with a strong accent. Even when the speaker pronounces several words in an unusual way, one can often correctly understand the sentence. We argue that the source of indirect supervision in processing unlabeled speech comes from prior knowledge about the world and the context of the speech.

Inspired by this, we devise a self-supervised learning framework termed *local prior matching* (LPM). We apply this framework to speech recognition allowing an ASR model to learn from unlabeled speech by leveraging a strong language model, which serves as the prior for self-supervision. Given an unlabeled utterance, the ASR model proposes multiple hypotheses and the language model provides a learning signal by evaluating the plausibility of each one.

We evaluate the LPM method on the LibriSpeech corpus (Panayotov et al., 2015), using 100 hours of labeled speech to seed the proposal model. Using 360 hours of additional *labeled* data reduces the word error rate (WER) by 3.8% and 13.1% absolute on an easier and a more-challenging test set, respectively. Using the same 360 hours but without labels, LPM reduces the WER by 2.1% and 9.5% absolute on the same two test sets, effectively bridging the gap to fully-supervised learning by 54% and 73%. In addition, by augmenting LPM with another 500 hours, for a total of 860 hours of unlabeled speech, LPM reduces WER on the noisier test set by 14.2% in total. Hence, LPM is able to surpass the performance of using 360 hours of labeled data by taking advantage of about twice the amount of unlabeled data. We also conduct extensive ablation studies and analyses in order to demonstrate the significance of each proposed component. For reproducibility, software will be open source and made publicly available in the camera-ready version.

## 2 METHOD

### 2.1 PRELIMINARIES

Let $\mathbf{x}$ denote an utterance of speech and $\mathbf{y}$ be a transcription. We assume speech is generated following a two-step process:

$$\mathbf{y} \sim p_{\mathbf{y}} \qquad \mathbf{x} \sim p_{\mathbf{x}|\mathbf{y}}, \tag{1}$$

where the text $\mathbf{y}$ is first generated from the language model (LM) $p_{\mathbf{y}}$, and the speech $\mathbf{x}$ is then generated from a text-to-speech (TTS) model $p_{\mathbf{x}|\mathbf{y}}$ conditioned on $\mathbf{y}$. The posterior $p_{\mathbf{y}|\mathbf{x}}$ is then the ASR model of interest. For convenience, we refer to $p_{\mathbf{xy}}(\mathbf{x}, \mathbf{y}) = p_{\mathbf{y}}(\mathbf{y}) \, p_{\mathbf{x}|\mathbf{y}}(\mathbf{x} \mid \mathbf{y})$ as the joint distribution, and $p_{\mathbf{x}}(\mathbf{x}) = \sum_{\mathbf{y}} p_{\mathbf{xy}}(\mathbf{x}, \mathbf{y})$ as the marginal distribution. In a typical supervised learning setting, one has access to a labeled dataset $\mathcal{D}_l$, which contains paired samples $\{(\boldsymbol{x}^{(i)}, \boldsymbol{y}^{(i)})\}_{i=1}^{N}$ drawn from the joint distribution. An ASR model $q_{\mathbf{y}|\mathbf{x}}$ can be trained by minimizing the marginal-weighted KL-divergence: $\mathbb{E}_{\mathbf{x} \sim p_{\mathbf{x}}}[D_{\mathrm{KL}}(p_{\mathbf{y}|\mathbf{x}} \mid\mid q_{\mathbf{y}|\mathbf{x}})]$, which can be estimated with $\sum_{(\boldsymbol{x}, \boldsymbol{y}) \in \mathcal{D}_l} -\log q_{\mathbf{y}|\mathbf{x}}(\boldsymbol{y} \mid \boldsymbol{x}) + const$ from samples (i.e. the commonly used cross-entropy loss). In the self-supervised setting we only have access to unlabeled speech dataset $\mathcal{D}_u^s$, yet we still wish to optimize the same marginal weighted KL-divergence. Since we do not have samples from $p_{\mathbf{y}|\mathbf{x}=\boldsymbol{x}}$ directly, we can, under mild assumptions, estimate it.

**Assumption 1** Different $\mathbf{y}$ can lead to the same pronunciation. One reason is the tokenization of words may not be unique. For example $\mathbf{y}$ can be the sequence of word-pieces "`c at_`" or "`ca t_`", both of which result in the same spoken word. Homophones (e.g. "right" and "write") also have the same pronunciation. To formalize this property, let $f : \mathbb{Y} \to \mathbb{Z}$ be a function which maps a text sequence to its pronunciation $\boldsymbol{z} \in \mathbb{Z}$ (a broader definition of "pronunciation" here, where $\boldsymbol{z} = f(\boldsymbol{y})$ is a distribution over phoneme sequences for a given text $\boldsymbol{y}$), and let $p_{\mathbf{x}|\mathbf{y}}(\mathbf{x} \mid \boldsymbol{y}) = p_{\mathbf{x}|\mathbf{z}}(\mathbf{x} \mid f(\boldsymbol{y}))$. We let the set $Y_{\boldsymbol{z}} = \{\boldsymbol{y} : f(\boldsymbol{y}) = \boldsymbol{z}\} \subset \mathbb{Y}$ contain all text samples with the same pronunciation and hence the same conditional distribution, $p_{\mathbf{x}|\mathbf{z}=\boldsymbol{z}}$.

**Assumption 2** We assume that two pairs $(\boldsymbol{y}, \boldsymbol{y}')$ that have different pronunciations (i.e., $f(\boldsymbol{y}) \neq f(\boldsymbol{y}')$) either sound similar or are very different. For instance, "lose" and "loose" result in almost the same pronunciation whereas an English speaker would rarely mistake "winter is coming" for "you know nothing". In the former case, the conditional distributions $p_{\mathbf{x}|\mathbf{y}}(\mathbf{x} \mid \boldsymbol{y})$ and $p_{\mathbf{x}|\mathbf{y}}(\mathbf{x} \mid \boldsymbol{y}')$ may overlap substantially whereas in the latter case they barely intersect. More formally, for each $\boldsymbol{z}$, we assume the pronunciation space $\mathbb{Z}$ can be partitioned by acoustic similarity as $Z_{\boldsymbol{z}}^{+} \cup Z_{\boldsymbol{z}}^{-}$. For all $\boldsymbol{z}^{+} \in Z_{\boldsymbol{z}}^{+}$, the conditional distributions are similar to that of $\boldsymbol{z}$, such that $D_{\mathrm{KL}}(p_{\mathbf{x}|\mathbf{z}=\boldsymbol{z}} \mid\mid p_{\mathbf{x}|\mathbf{z}=\boldsymbol{z}^{+}}) \leq \epsilon$ for small $\epsilon$. On the other hand, we assume the set of dissimilar pronunciations $Z_{\boldsymbol{z}}^{-}$ contributes very little to the marginal probability of non-outlier speech samples. In other words, for any $\boldsymbol{z}$ and $\boldsymbol{x}$ where $p_{\mathbf{x}|\mathbf{z}}(\boldsymbol{x} \mid \boldsymbol{z}) \geq \delta$, we assume $\sum_{\boldsymbol{z}^{-} \in Z_{\boldsymbol{z}}^{-}} \sum_{\boldsymbol{y}^{-} \in Y_{\boldsymbol{z}^{-}}} p_{\mathbf{x}|\mathbf{y}}(\boldsymbol{x} \mid \boldsymbol{y}^{-}) \, p_{\mathbf{y}}(\boldsymbol{y}^{-}) \ll p_{\mathbf{x}}(\boldsymbol{x})$.

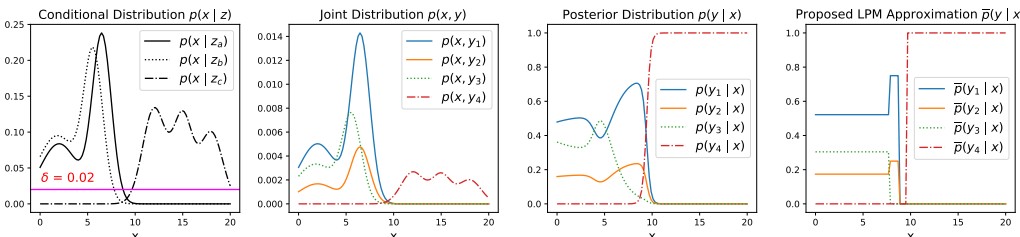

Figure 1: Illustration of the assumed generative process and the proposed posterior approximation. The leftmost figure shows $p_{\mathbf{x}|\mathbf{z}}$ of three pronunciations: $\boldsymbol{z}_a$, $\boldsymbol{z}_b$ and $\boldsymbol{z}_c$, where $\boldsymbol{z}_a, \boldsymbol{z}_b \in Z_{\boldsymbol{z}_a}^{+}$ and $\boldsymbol{z}_c \in Z_{\boldsymbol{z}_a}^{-}$. The following two figures depict the joint and the posterior distributions of four texts, respectively, where it is assumed $f(\boldsymbol{y}_1) = f(\boldsymbol{y}_2) = \boldsymbol{z}_a$, $f(\boldsymbol{y}_3) = \boldsymbol{z}_b$, $f(\boldsymbol{y}_4) = \boldsymbol{z}_c$, and $p_{\mathbf{y}}(\boldsymbol{y}_1) > p_{\mathbf{y}}(\boldsymbol{y}_3) > p_{\mathbf{y}}(\boldsymbol{y}_2) > p_{\mathbf{y}}(\boldsymbol{y}_4)$. The rightmost figure presents *local prior*, our proposed approximation to the $p_{\mathbf{y}|\mathbf{x}}$, computed from the priors of texts whose $p_{\mathbf{x}|\mathbf{y}}$ are above the threshold $\delta$ at a given $\boldsymbol{x}$.

## 2.2 LOCAL PRIOR MATCHING

Equipped with the above assumptions, we next present a tractable approximation of the posterior $p_{\mathbf{y}|\mathbf{x}}$ using unlabeled speech data. Figure 1 shows a toy example of the proposed approach. Let $\boldsymbol{x}$ be a non-outlier speech sample that is generated conditioned on a text sample $\boldsymbol{y}^*$, whose pronunciation is $f(\boldsymbol{y}^*) = \boldsymbol{z}^*$. The posterior can be derived with Bayes' rule as follows:

$$p_{\mathbf{y}|\mathbf{x}}(\boldsymbol{y} \mid \boldsymbol{x}) = \frac{p_{\mathbf{y}}(\boldsymbol{y}) \, p_{\mathbf{x}|\mathbf{y}}(\boldsymbol{x} \mid \boldsymbol{y})}{\sum_{\hat{\boldsymbol{y}} \in \mathbb{Y}} p_{\mathbf{y}}(\hat{\boldsymbol{y}}) \, p_{\mathbf{x}|\mathbf{y}}(\boldsymbol{x} \mid \hat{\boldsymbol{y}})} = \frac{p_{\mathbf{y}}(\boldsymbol{y}) \, p_{\mathbf{x}|\mathbf{y}}(\boldsymbol{x} \mid \boldsymbol{y})}{\sum_{\hat{\boldsymbol{z}} \in Z_{\boldsymbol{z}^*}^+ \cup Z_{\boldsymbol{z}^*}^-} \sum_{\hat{\boldsymbol{y}} \in Y_{\hat{\boldsymbol{z}}}} p_{\mathbf{y}}(\hat{\boldsymbol{y}}) \, p_{\mathbf{x}|\mathbf{y}}(\boldsymbol{x} \mid \hat{\boldsymbol{y}})} \tag{2}$$

$$\approx \frac{p_{\mathbf{y}}(\boldsymbol{y}) \, p_{\mathbf{x}|\mathbf{y}}(\boldsymbol{x} \mid \boldsymbol{y})}{\sum_{\hat{\boldsymbol{z}} \in Z_{\boldsymbol{z}^*}^+} \sum_{\hat{\boldsymbol{y}} \in Y_{\hat{\boldsymbol{z}}}} p_{\mathbf{y}}(\hat{\boldsymbol{y}}) \, p_{\mathbf{x}|\mathbf{y}}(\boldsymbol{x} \mid \hat{\boldsymbol{y}})} \cdot \mathbb{1}_{\cup_{\hat{\boldsymbol{z}} \in Z_{\boldsymbol{z}^*}^+} Y_{\hat{\boldsymbol{z}}}}(\boldsymbol{y}) \tag{3}$$

$$= \frac{p_{\mathbf{y}}(\boldsymbol{y}) \, p_{\mathbf{x}|\mathbf{z}}(\boldsymbol{x} \mid f(\boldsymbol{y}))}{\sum_{\hat{\boldsymbol{z}} \in Z_{\boldsymbol{z}^*}^+} \sum_{\hat{\boldsymbol{y}} \in Y_{\hat{\boldsymbol{z}}}} p_{\mathbf{y}}(\hat{\boldsymbol{y}}) \, p_{\mathbf{x}|\mathbf{z}}(\boldsymbol{x} \mid \hat{\boldsymbol{z}})} \cdot \mathbb{1}_{\cup_{\hat{\boldsymbol{z}} \in Z_{\boldsymbol{z}^*}^+} Y_{\hat{\boldsymbol{z}}}}(\boldsymbol{y}). \tag{4}$$

The set $\mathbb{Y}$ can be partitioned according to pronunciation to arrive at equation 2, which is then approximated using by remark 2 to get equation 3, where $\mathbb{1}$ is the indicator function. In equation 4, $p_{\mathbf{x}|\mathbf{y}}(\boldsymbol{x} \mid \hat{\boldsymbol{y}})$ is simply replaced with $p_{\mathbf{x}|\mathbf{z}}(\boldsymbol{x} \mid f(\hat{\boldsymbol{y}}))$ as stated in remark 1. Despite the simplification, estimating equation 4 requires a language model, a TTS model, and the enumeration of text sequences with similar pronunciations to that of $\boldsymbol{z}^*$. To avoid the latter two requirements, we make the following approximations:

1. Replace $p_{\mathbf{x}|\mathbf{z}}(\boldsymbol{x} \mid \hat{\boldsymbol{z}})$ with $p_{\mathbf{x}|\mathbf{z}}(\boldsymbol{x} \mid \boldsymbol{z}^*)$ for all $\hat{\boldsymbol{z}} \in Z_{\boldsymbol{z}^*}^+$ by exploiting the assumption that these distributions are very similar to each other (i.e., from remark 2 we have $D_{\mathrm{KL}}(p_{\mathbf{x}|\mathbf{z}=\boldsymbol{z}^*} \parallel p_{\mathbf{x}|\mathbf{z}=\hat{\boldsymbol{z}}}) \le \epsilon$).
2. Replace $\cup_{\hat{\boldsymbol{z}} \in Z_{\boldsymbol{z}^*}^+} Y_{\hat{\boldsymbol{z}}}$ with hypotheses proposed from a beam search using a bootstrapped ASR proposal model. We refer to the proposal model as $r_{\mathbf{y}|\mathbf{x}}$ and the set of hypotheses for $\boldsymbol{x}$ generated from the beam search as $B(r_{\mathbf{y}|\mathbf{x}}, \boldsymbol{x}, k)$, where $k$ is the beam size.

The approximated posterior $\bar{p}_{\mathbf{y}|\mathbf{x}}(\boldsymbol{y} \mid \boldsymbol{x})$ can then be written as:

$$\bar{p}_{\mathbf{y}|\mathbf{x}}(\boldsymbol{y} \mid \boldsymbol{x}) = \frac{p_{\mathbf{y}}(\boldsymbol{y}) \, p_{\mathbf{x}|\mathbf{z}}(\boldsymbol{x} \mid \boldsymbol{z}^*)}{\sum_{\hat{\boldsymbol{y}} \in B(r_{\mathbf{y}|\mathbf{x}}, \boldsymbol{x}, k)} p_{\mathbf{y}}(\hat{\boldsymbol{y}}) \, p_{\mathbf{x}|\mathbf{z}}(\boldsymbol{x} \mid \boldsymbol{z}^*)} \cdot \mathbb{1}_{B(r_{\mathbf{y}|\mathbf{x}}, \boldsymbol{x}, k)}(\boldsymbol{y}) \tag{5}$$

$$= \frac{p_{\mathbf{y}}(\boldsymbol{y})}{\sum_{\hat{\boldsymbol{y}} \in B(r_{\mathbf{y}|\mathbf{x}}, \boldsymbol{x}, k)} p_{\mathbf{y}}(\hat{\boldsymbol{y}})} \cdot \mathbb{1}_{B(r_{\mathbf{y}|\mathbf{x}}, \boldsymbol{x}, k)}(\boldsymbol{y}). \tag{6}$$

The approximated posterior only requires computing language model probabilities of the beam search hypotheses, which is tractable. We refer to equation 6 as the *local prior*, since it is a re-normalized distribution in the neighborhood of $\boldsymbol{y}^*$. We also propose *local prior matching* (LPM) as an unsupervised objective for training an ASR model $q_{\mathbf{y}|\mathbf{x}}$:

$$\mathcal{L}_{lpm}(q_{\mathbf{y}|\mathbf{x}}; \boldsymbol{x}, p_{\mathbf{y}}, r_{\mathbf{y}|\mathbf{x}}, k) = D_{\mathrm{KL}}(\bar{p}_{\mathbf{y}|\mathbf{x}=\boldsymbol{x}} \| q_{\mathbf{y}|\mathbf{x}=\boldsymbol{x}})$$

$$= -H(\bar{p}_{\mathbf{y}|\mathbf{x}=\boldsymbol{x}}) - \sum_{\boldsymbol{y} \in B(r_{\mathbf{y}|\mathbf{x}}, \boldsymbol{x}, k)} \frac{p_{\mathbf{y}}(\boldsymbol{y})}{\sum_{\hat{\boldsymbol{y}} \in B(r_{\mathbf{y}|\mathbf{x}}, \boldsymbol{x}, k)} p_{\mathbf{y}}(\hat{\boldsymbol{y}})} \log q_{\mathbf{y}|\mathbf{x}}(\boldsymbol{y} \mid \boldsymbol{x}),$$

which minimizes the KL-divergence between the local prior and the model distribution. For clarity, we term $q_{\mathbf{y}|\mathbf{x}}$ the *online model*, and let $q_{\mathbf{y}|\mathbf{x}}(\boldsymbol{y} \mid \mathbf{x}; \theta_q)$ and $r_{\mathbf{y}|\mathbf{x}}(\boldsymbol{y} \mid \mathbf{x}; \theta_r)$ denote the models that are parameterized by and $\theta_q$ and $\theta_r$, respectively. Since the LPM objective simply re-weights hypotheses by the local prior, we do not need a continuous relaxation of $\mathbf{y}$ to make $p_{\mathbf{y}}$ differentiable with respect to $\theta_q$ as in Yang et al. (2018); Liu et al. (2019). The gradient of the proposed LPM objective is given by:

$$\frac{\partial \mathcal{L}_{lpm}(q_{\mathbf{y}|\mathbf{x}}; \boldsymbol{x}, p_{\mathbf{y}}, r_{\mathbf{y}|\mathbf{x}}, k)}{\partial \theta_q} = \sum_{\boldsymbol{y} \in B(r_{\mathbf{y}|\mathbf{x}}, \boldsymbol{x}, k)} \frac{p_{\mathbf{y}}(\boldsymbol{y})}{\sum_{\hat{\boldsymbol{y}} \in B(r_{\mathbf{y}|\mathbf{x}}, \boldsymbol{x}, k)} p_{\mathbf{y}}(\hat{\boldsymbol{y}})} \frac{\partial \log q_{\mathbf{y}|\mathbf{x}}(\boldsymbol{y} \mid \boldsymbol{x})}{\partial \theta_q}.$$

We use paired audio and text to estimate $r_{\mathbf{y}|\mathbf{x}}$ and unpaired speech to compute the LPM objective. The LPM objective can also take advantage of large unpaired text corpora in order to obtain better language models which directly improve the quality of the approximate posterior.

## 2.3 PROPOSAL MODEL

The quality of the posterior approximation $\bar{p}_{\mathbf{y}|\mathbf{x}}$ depends on the proposal model $r_{\mathbf{y}|\mathbf{x}}$. Instead of using a fixed proposal model throughout the entire training process, we update $r_{\mathbf{y}|\mathbf{x}}$ with $q_{\mathbf{y}|\mathbf{x}}$ if the latter is better.

**On-policy beam search** The first approach always uses the online model $q_{\mathbf{y}|\mathbf{x}}$ as the proposal model. This means $\theta_r = \theta_q$ and is effectively a form of *on-policy beam search*, since the model used to generate hypotheses is also the model we update.

**Off-policy beam search** While the on-policy method benefits from the immediate improvement of the online model, it also suffers immediately if the gradient update from a mini-batch deteriorates performance. This can result in instability during optimization. We consider a second option which does not tie $\theta_r$ and $\theta_q$ but instead updates $\theta_r$ with $\theta_q$ every $T$ steps only when the performance of the online model $q_{\mathbf{y}|\mathbf{x}}$ is better than that of the proposal model $r_{\mathbf{y}|\mathbf{x}}$ by some metric. We refer to the second option as *off-policy beam search*. To avoid overfitting to the training set, we use the character error rate (CER) on the validation set as the metric for the proposal model update. We set $T = 1000$ for all experiments with the off-policy beam search.

## 2.4 FILTERING HYPOTHESES USING ESTIMATED LENGTHS

As noted in Chorowski & Jaitly (2017), sequence-to-sequence ASR models sometimes predict end-of-sentence (EOS) tokens too early or generate looping n-grams, resulting in hypotheses that are significantly shorter or longer than the set of acoustically matched texts for a given utterance. Of the two failure modes, the former is more harmful to training with LPM. The reason is that the LPM objective assumes all hypotheses obtained from beam search are acoustically reasonable, and weights each of them by linguistic plausibility given by an LM. While LMs are effective in discriminating plausibility between sentences of similar lengths, we find empirically they tend to assign higher probabilities to shorter sentences than to longer sentences, even when the longer ones are more plausible and grammatically correct than the shorter ones. As a result, truncated hypotheses are assigned higher weights than those that are acoustically matched but longer, which in turn encourages earlier prediction of EOS tokens and forms a catastrophic feedback loop particularly with the on-policy beam search.

To address this issue, we propose a simple filtering heuristic based on the text length. Before training the model, a text length $L$ is estimated for each unlabeled speech sample $\boldsymbol{x}$. During training, only hypotheses with length close to $L$ are retained for the LPM objective computation. Let $len(\boldsymbol{y})$ denote the length of $\boldsymbol{y}$. We keep a hypothesis $\boldsymbol{y}$ only if $\lfloor r_{lb} \cdot L \rfloor \leq len(\boldsymbol{y}) \leq \lceil r_{ub} \cdot L \rceil$, where $r_{lb}$ and $r_{ub}$ are the text length lower and upper bound ratios, respectively. Several methods can be used to estimate the text length on an unlabeled utterance, including using the average speaking rate (Peng et al., 2019) or generating a phoneme/syllable segmentation (Adell & Bonafonte, 2004; Scharenborg et al., 2010; Wang et al., 2017). In this work, we estimate the length by using using that of the best hypothesis generated from the initial proposal model, generated with either ASR-only greedy decoding or ASR+LM beam search decoding.

## 3 EXPERIMENTAL SETUP

**Dataset** We evaluate our approach on LibriSpeech (Panayotov et al., 2015), a crowd-sourced audio book corpus derived from the LibriVox Project. The training set contains 960 hours of speech, officially split into three sets: train-clean-100, train-clean-360, and train-other-500, where the first two sets are easier and the third set is noisier and more accented. Similarly, the development and test sets are also split according to difficulty, resulting in four partitions: {dev, test}×{clean, other}, each of which contains roughly five hours of speech. In this work we use train-clean-100 as the paired speech data, and the other two training splits as the unpaired data.

We train the language model on the unpaired text data provided with LibriSpeech, which includes approximately 14,500 books collected from Project Gutenberg. Some of the books in the text corpus overlap with those in the LibriSpeech training set. To avoid training the LM on the ground truth text of the unlabeled speech, we exclude the 997 overlapping books from the text data. We follow the same recipe as Kahn et al. (2019) to pre-process the remaining text.

**Neural Network Architecture** The proposal model $r_{\mathbf{y}|\mathbf{x}}$ and the online model $q_{\mathbf{y}|\mathbf{x}}$ are sequence-to-sequence neural networks (Bahdanau et al., 2016; Chorowski & Jaitly, 2017) with the same time-depth separable (TDS) architecture proposed in Hannun et al. (2019). The encoder is fully convolutional, composed of TDS blocks which reduce the number of parameters while keeping the receptive field large. The decoder is a single layer recurrent neural network (RNN) with gated recurrent units (GRUs), equipped with a single-headed inner-product key-value attention (Vaswani et al., 2017) for querying information from the encoder outputs. We follow the recipe of Kahn et al. (2019) which uses fewer TDS blocks in the encoder compared to Hannun et al. (2019) in order to generalize better when trained on the smaller LibriSpeech train-clean-100. The output target of the decoder at each step is a posterior distribution over 5,000 word pieces, generated with the SentencePiece toolkit (Kudo & Richardson, 2018) using transcripts from train-clean-100.

To enable efficient evaluation of the language model probabilities, which is required at each training step, we use the gated convolutional language model architecture (ConvLM) proposed in Dauphin et al. (2017), which achieves competitive performances compared to recurrent models while significantly reducing the latency. We use the same 5,000 word-piece vocabulary for the LM which is trained with the same model configuration and recipe as Zeghidour et al. (2018). The trained ConvLM achieves a token perplexity of 34.24 on the development set.

**Optimization** We use both paired and unpaired data to optimize $q_{\mathbf{y}|\mathbf{x}}$. To simplify the optimization procedure, the model is provided with either a paired or an unpaired batch at each step, alternated with a fixed ratio $m_l : m_u$. When given a paired batch of $n$ samples, $\{(\boldsymbol{x}^{(i)}, \boldsymbol{y}^{(i)})\}_{i=1}^{n}$, the model minimizes the standard cross-entropy loss, $\frac{1}{n} \sum_i - \log q_{\mathbf{y}|\mathbf{x}}(\boldsymbol{y}^{(i)} \mid \boldsymbol{x}^{(i)})$. When provided with an unpaired batch $\{\boldsymbol{x}^{(i)}\}_{i=1}^{n}$, the model minimizes a weighted LPM loss, $\frac{\alpha}{n} \cdot \sum_i \mathcal{L}_{lpm}(q_{\mathbf{y}|\mathbf{x}}; \boldsymbol{x}^{(i)}, p_{\mathbf{y}}, r_{\mathbf{y}|\mathbf{x}}, k)$. The weight $\alpha$ and the mixing ratio $m_l/m_u$ are used to balance the supervised and self-training objectives. For regularization we use 20% dropout (Srivastava et al., 2014), 10% label smoothing, 1% decoder input sampling, and 1% word piece sampling (Kudo, 2018) following Kahn et al. (2019). We use SGD without momentum to train the online model with an initial learning rate of 5e-2. To achieve a good CER on the development sets, the model is trained for at least 1.6M steps (paired and unpaired ones combined) with a batch size of 16 (8 GPUs × 2 per GPU). The learning rate is annealed by a factor of two every 0.64M steps. All experiments in this paper are implemented in the wav2letter++ framework (Pratap et al., 2018).

**Initialization** To initialize the proposal model and the online model, we consider three checkpoints from a baseline model trained on train-clean-100 for a varying number of steps using only the supervised objective. The three checkpoints, whose parameters are denoted as $\theta_A$, $\theta_B$, and $\theta_C$, are trained for about 300k / 40k / 16k steps, achieving average development set CERs of 13% / 20% / 38%, respectively. To obtain a better approximation $\bar{p}_{\mathbf{y}|\mathbf{x}}$ for the posterior, it is desirable to use the best-performed checkpoint to initialize the proposal model. By contrast, as observed in Kahn et al. (2019) that training from scratch achieves consistently better performances than starting from a well-trained model, we hypothesize that initializing the online model with an earlier checkpoint may also have a similar effect. Hence, we initialize $\theta_r = \theta_A$ and $\theta_q = \theta_C$ if not otherwise specified.

## 4 RESULTS

The best supervised model trained only on train-clean-100 ($\theta_A$) achieves a WER of 14.00%/37.02% on dev-clean/dev-other, respectively. Unless otherwise stated, we use train-clean-360 as the unpaired speech dataset, $(r_{lb}, r_{ub}) = (0.95, 1.05)$ for length filtering with reference lengths obtained from ASR-only greedy decoding.

### 4.1 BEAM SIZE, MIXING RATIO, AND LPM WEIGHTS

Table 1 shows how the WER varies with the beam size and the mixing ratio. for all mixing ratios, the model improves the most from beam size of $k = 1$ to $k = 2$, showing the benefit of considering multiple hypotheses. The improvement is greater when a higher mixing ratio of unpaired-to-paired speech is used. In addition, we note that the LM is effectively unused when $k = 1$ because the LM probability assigned to each hypothesis is normalized within the beam. If there is only one hypothesis, it will be assigned an approximate posterior probability of one. The amount of improvement

Table 1: Vary mixing ratio and beam size. An LPM weight $\alpha = 0.2$ is used.

| $\mathcal{D}_u^s$ | $r_l : r_u$ | dev-clean / dev-other WER (%) | | | | |
|---|---|---|---|---|---|---|
| | | $k = 1$ | $k = 2$ | $k = 4$ | $k = 8$ | $k = 16$ |
| 360hr | 4 : 1 | 10.75 / 31.62 | 10.60 / 30.96 | 10.25 / 30.67 | 10.14 / 30.17 | 10.09 / 29.99 |
| | 1 : 1 | 10.43 / 29.76 | 9.56 / 28.83 | 9.37 / 28.10 | 9.06 / 27.35 | 8.88 / 27.25 |
| | 1 : 4 | 11.09 / 29.89 | 9.34 / 27.45 | **9.00 / 26.47** | 9.15 / 26.52 | 9.36 / 27.00 |
| | 1 : 9 | 12.11 / 30.89 | 10.11 / 27.71 | 9.76 / 27.08 | 10.17 / 27.41 | 10.30 / 27.62 |
| 860hr | 1 : 4 | 10.59 / 26.05 | 9.37 / 23.85 | 8.68 / 22.53 | 8.37 / 21.56 | **8.37 / 21.33** |

diminishes with larger beam sizes, and the performance even starts to degrade beyond $k = 4$ when using a higher mixing ratio. This may result from the inclusion of worse hypotheses which have a better scored under the LM. We use the best setting for the following experiments with a mixing ratio $r_l : r_u = 1 : 4$, a beam size $k = 4$, and an LPM weight $\alpha = 0.2$. We present detailed results varying the LPM weight in the Appendix.

## 4.2 PROPOSAL MODEL UPDATE AND MODEL INITIALIZATION

In addition to the two proposal model update strategies proposed in Section 2.3, termed *on-policy* and *off-policy (better)*, we experiment with two additional strategies. The first, *off-policy (never)*, uses a fixed proposal model throughout training without ever updating. The second, *off-policy (always)*, updates the proposal model with the online model every $T$ steps (i.e., set $\theta_r \leftarrow \theta_q$) regardless of the performance.

The full results are shown in Table 2. Four key takeaways are as follows. (1) For all combinations of $(r_{\mathbf{y}|\mathbf{x}}, q_{\mathbf{y}|\mathbf{x}})$ initialization, off-policy (never) is the worst. This demonstrates the importance of updating the proposal model to generate better hypotheses during training. (2) Off-policy (always) consistently outperforms on-policy. We observe that training is significantly stabilized by reducing the proposal model update frequency from every step to every 1,000 steps. The effect is particularly prominent when initializing $r_{\mathbf{y}|\mathbf{x}}$ and $q_{\mathbf{y}|\mathbf{x}}$ from an earlier checkpoint (9.62% vs 20.02% on dev-clean, and 27.51% vs 45.62% on dev-other). (3) Off-policy (better) achieves the best WER in all settings and outperforms off-policy (always) by a larger margin when initializing from an earlier checkpoint. (4) Unlike the other strategies, off-policy (better) demonstrates consistent improvement when using a less-trained initial online model. In the following experiments, we initialize models with $\theta_r = \theta_A$ and $\theta_q = \theta_C$ unless otherwise specified.

Table 2: Vary proposal model update strategies and online model initialization.

| Init $r_{\mathbf{y}|\mathbf{x}}$ | $r_{\mathbf{y}|\mathbf{x}}$ update | dev-clean / dev-other WERs (%) | | |
|---|---|---|---|---|
| | | Init $q_{\mathbf{y}|\mathbf{x}} = A$ | Init $q_{\mathbf{y}|\mathbf{x}} = B$ | Init $q_{\mathbf{y}|\mathbf{x}} = C$ |
| A | On-Policy | 9.50 / 28.29 | N/A | N/A |
| | Off-Policy (never) | 11.19 / 31.74 | 11.14 / 31.69 | 11.24 / 31.53 |
| | Off-Policy (always) | 9.40 / 27.79 | 9.27 / 27.33 | 9.52 / 27.34 |
| | Off-Policy (better) | 9.20 / 27.42 | 9.14 / 26.80 | **9.00 / 26.47** |
| B | On-Policy | N/A | 10.17 / 28.35 | N/A |
| | Off-Policy (never) | 13.61 / 35.39 | 13.95 / 35.43 | 13.56 / 35.62 |
| | Off-Policy (always) | 9.50 / 27.81 | 9.56 / 27.58 | 9.79 / 27.44 |
| | Off-Policy (better) | 9.30 / 27.34 | 9.26 / 27.01 | **9.15 / 26.63** |
| C | On-Policy | N/A | N/A | 20.20 / 45.62 |
| | Off-Policy (never) | 20.59 / 44.09 | 22.95 / 46.43 | 23.42 / 46.89 |
| | Off-Policy (always) | 9.52 / 27.89 | 9.46 / 27.35 | 9.62 / 27.51 |
| | Off-Policy (better) | 9.44 / 27.34 | **9.31** / 27.26 | 9.43 / **27.19** |

### 4.3 LENGTH FILTERING

Table 3 shows the impact of length filtering with different proposal model update strategies, where both the proposal model and the online model are initialized with $\theta_A$. As discussed in Section 2.4, on-policy suffers more than off-policy without length filtering. If the proposal model is never updated, then length filtering does not affect the final WER. We hypothesize that length filtering keeps the proposal model stable during training. Table 4 compares three different reference lengths: the oracle length (*Oracle*), the predicted length from ASR + LM beam search decoding (*ASR + LM Dec*), and the predicted length from ASR-only greedy decoding (*ASR-only Dec*). We observe that the WER does not differ much when using different reference lengths for filtering.

Table 3: Ablation study for length filtering.

| $r_{\mathbf{y}|\mathbf{x}}$ update | dev-clean / dev-other WER (%) | |
| | No filtering | With filtering |
|---|---|---|
| On-Policy | 26.65 / 59.07 | 9.50 / 28.29 |
| Off-Policy (never) | 11.18 / 31.83 | 11.19 / 31.74 |
| Off-Policy (always) | 13.99 / 35.52 | 9.40 / 27.79 |
| Off-Policy (better) | 11.42 / 31.56 | **9.20 / 27.42** |

Table 4: Length filtering using different reference lengths.

| Reference Length $L$ | dev-{clean / other} |
|---|---|
| Oracle | 8.85 / 26.39 |
| ASR + LM Dec | 8.99 / 26.36 |
| ASR-only Dec | 9.00 / 26.47 |

### 4.4 CHOICE OF LANGUAGE MODELS

We study how the LM affects the performance by experimenting with three alternative hypothesis weighting methods. The first method uses the LM score normalized by token length, $p_{\mathbf{y}}(\boldsymbol{y})^{1/len(\boldsymbol{y})}$, which avoids favoring overly short hypotheses. The second method computes the LM probability of each hypothesis and shuffles them, which preserves the entropy of the original approximate posterior but randomizes the weights. The third assigns an equal weight for each hypothesis. Table 5 presents the results with three beam sizes: $k = 2, 4, 8$. Using the LM weights directly performs the best for all beam sizes. Length normalization (LenNorm) is better than shuffled and uniform, which have similar performance. This suggests that both ranking hypotheses correctly and scaling the weights proportional to the LM probability are important.

We also observe how the quality of the LM affects the results. We test LMs of various quality by using the same ConvLM but trained for a different number of steps. We quantify LM quality by the token perplexity (PPL) on the development set. Table 6 shows a clear positive correlation between the LM quality and the final WER. This is expected given that the better LM results in a more accurate posterior approximation $\bar{p}_{\mathbf{y}|\mathbf{x}}$.

Table 5: Comparison of LPM and alternative hypothesis weighting methods.

| $\bar{p}_{\mathbf{y}|\mathbf{x}}$ | dev-clean / dev-other WER (%) | | |
| | $k = 2$ | $k = 4$ | $k = 8$ |
|---|---|---|---|
| Proposed | **9.34 / 27.45** | **9.00 / 26.47** | **9.15 / 26.52** |
| LenNorm | 10.52 / 29.10 | 10.32 / 28.69 | 10.20 / 28.39 |
| Shuffled | 10.76 / 29.32 | 10.94 / 29.17 | 10.56 / 29.20 |
| Uniform | 10.96 / 29.46 | 10.74 / 29.09 | 10.84 / 29.21 |

Table 6: Using $p_{\mathbf{y}}$ of different development token perplexity.

| $p_{\mathbf{y}}$ PPL | dev-{clean / other} |
|---|---|
| 34.24 | **9.00 / 26.47** |
| 64.22 | 10.08 / 26.92 |
| 97.87 | 10.90 / 27.97 |
| 142.12 | 11.53 / 28.74 |
| 180.71 | 13.18 / 30.74 |

### 4.5 FINAL RESULTS AND COMPARISON WITH LITERATURE

The best performing model is trained for 3.2M steps, with a learning rate annealed by a factor of two every 1.28M steps when using 360 hours of unpaired speech, and every 0.64M steps when using 860 hours of unpaired speech. Reference lengths for filtering are obtained from ASR+LM beam search decoding. We compare LPM to fully supervised models and a pseudo labeling (PL) method in Table 7. Compared to LPM, PL demands extensive hyper-parameter tuning for the ASR+LM beam search as well as heuristic data filtering methods. to obtain high quality pseudo transcripts. We

chose PL as a baseline because we follow the same experimental setup, and, more importantly, it achieves the state-of-the-art results on LibriSpeech when using train-clean-100 as paired data and train-other-360 as unpaired speech. A more complete table with results from the literature is in the Appendix.

The upper half of Table 7 shows greedy decoding results without an LM. For the fully supervised model, when removing train-clean-360 the WER increases by 6.86% on test-clean and 13.36% on test-other. Using train-clean-360 speech without transcripts, LPM reduces the absolute WER by 5.64% and 12.21% on the two test sets, which recovers 82% and 91%, respectively, of the WER drop from removing the labels. Adding noisier train-other-500 to the unpaired set (total 860hr $\mathcal{D}_u^s$) further reduces the WER, and LPM achieves a better WER on the noisy sets (dev-other and test-other) compared to the supervised model trained on 460 hours of clean paired data. In addition, LPM outperforms PL in all settings. This trend is consistent even when decoding with a strong ConvLM.

Table 7: Results of baselines and the proposed methods.

| | $\mathcal{D}_l$ | $\mathcal{D}_u^s$ | LM | dev WER (%) | | test WER (%) | |
| --- | --- | --- | --- | --- | --- | --- | --- |
| | | | | clean | other | clean | other |
| Supervised | 100hr | N/A | None | 14.00 | 37.02 | 14.85 | 39.95 |
| Supervised | 460hr | N/A | None | 7.20 | 25.32 | 7.99 | 26.59 |
| PL (ASR greedy) (Kahn et al., 2019) | 100hr | 360hr | None | 12.27 | 33.42 | 12.57 | 35.36 |
| PL (ASR+LM stable) (Kahn et al., 2019) | 100hr | 360hr | None | 9.30 | 28.79 | 9.84 | 30.15 |
| PL (ASR+LM stable) (Kahn et al., 2019) | 100hr | 860hr | None | 9.03 | 26.03 | 9.44 | 27.25 |
| Local Prior Matching | 100hr | 360hr | None | 8.85 | 26.33 | 9.21 | 27.74 |
| Local Prior Matching | 100hr | 860hr | None | **8.08** | **21.52** | **8.37** | **22.89** |
| Supervised | 100hr | N/A | ConvLM | 7.78 | 28.15 | 8.06 | 30.44 |
| Supervised | 460hr | N/A | ConvLM | 3.98 | 17.00 | 4.23 | 17.36 |
| PL (ASR greedy) (Kahn et al., 2019) | 100hr | 360hr | ConvLM | 6.19 | 23.53 | 6.81 | 24.99 |
| PL (ASR+LM stable) (Kahn et al., 2019) | 100hr | 360hr | ConvLM | 5.73 | 22.54 | 6.35 | 24.13 |
| PL (ASR+LM stable) (Kahn et al., 2019) | 100hr | 860hr | ConvLM | 6.31 | 21.87 | 6.84 | 23.29 |
| Local Prior Matching | 100hr | 360hr | ConvLM | 5.69 | 20.22 | 5.99 | 20.93 |
| Local Prior Matching | 100hr | 860hr | ConvLM | **5.39** | **14.89** | **5.78** | **16.27** |

## 5 ANALYSIS

### 5.1 HYPOTHESIS QUALITY OF UNLABELED TRAINING SPEECH

As discussed in Section 4.2, updating the proposal model is crucial to improve the hypotheses used during training. To quantify the improvement, Table 8 shows WERs on the unlabeled data of an LPM model at the beginning (step=0) and at the end (step=3.2M) of training. Note that this is a proxy of quality for LPM, since multiple hypotheses are used when setting $k \geq 2$. We compare to the WER of the pseudo transcripts used in PL. Although generating hypotheses without an LM leads to worse WER than that of PL, toward the end of training, the proposal model of LPM produces better predictions on both train-clean-360 and train-other-500 than the fixed ones used in PL (ASR+LM stable). Furthermore, the WER on train-other-500 much higher for PL (21.51%) than for LPM at the end of training (13.00%), which explains why LPM achieves much better performances than PL when using the full 860hr of unpaired data.

### 5.2 LINGUISTIC PLAUSIBILITY

We expect models trained with LPM to generate more semantically and grammatically correct text since the ASR model receives direct supervision from the LM. Table 9 shows the proposed hypotheses for two utterances using a supervised baseline model and a model trained with LPM. The baseline model proposes erroneous hypotheses which are easy to discard even without the audio or the ground truth transcript. On the other hand, LPM generates hypotheses that are both grammatically and semantically plausible, with acceptable substitution errors in some cases (e.g., might/would).

Table 8: Comparison of the label quality on the unlabeled speech data between the two methods when trained on 100hr $\mathcal{D}_l$ and 860hr $\mathcal{D}_u^s$.

| Labelling Method | | Step | train WER (%) | |
| --- | --- | --- | --- | --- |
| | | | clean-360 | other-500 |
| Pseudo Label | ASR+LM stable | All | 8.25 | 21.51 |
| Local Prior Match | Proposal greedy | 0 | 14.81 | 29.03 |
| Local Prior Match | Proposal greedy | 3.2M | 7.37 | 13.00 |

Table 9: Comparison of beam search hypotheses from the supervised model (100hr $\mathcal{D}_l$) and the model trained with LPM (100hr $\mathcal{D}_l$ + 360hr $\mathcal{D}_u^s$). Hypotheses are ranked by ASR score $q_{\mathbf{y}|\mathbf{x}}$.

| Model | Rank | $\log p_{\mathbf{y}}$ | Beam Search Hypotheses ($k = 4$) |
| --- | --- | --- | --- |
| Ref. | - | -34.14 | _she _walk ed _very _fast _after _she _left _the _house $ |
| Sup. | 1 | -47.58 | _she _looked _very _thought _after _she _left _the _house $ |
| | 2 | -38.54 | _she _looked _very _fat _after _she _left _the _house $ |
| | 3 | -53.47 | _she _what _very _thought _after _she _left _the _house $ |
| | 4 | -83.85 | _she _ w o u l t _very _thought _after _she _left _the _house $ |
| LPM | 1 | -32.24 | _she _walk ed _very _fast _as _she _left _the _house $ |
| | 2 | -34.14 | _she _walk ed _very _fast _after _she _left _the _house $ |
| | 3 | -33.10 | _she _looked _very _fast _as _she _left _the _house $ |
| | 4 | -36.59 | _she _looked _very _fast _after _she _left _the _house $ |
| Ref. | - | -68.01 | _oh _if _i _had _imagined _him _still _in _such _distress _sure ly _i _might _have _done _something _to _help _him $ |
| Sup. | 1 | -110.11 | _i _before _i _had _imagined _him _steal ing _such _distress _sure ly _i _why _have _done _something _to _help _you $ |
| | 2 | -107.55 | _i _before _i _had _imagined _him _steal ing _such _distress _sure ly _i _want _have _done _something _to _help _you $ |
| | 3 | -107.81 | _i _before _i _had _imagined _him _still ing _such _distress _sure ly _i _want _have _done _something _to _help _you $ |
| | 4 | -107.10 | _i _before _i _had _imagined _him _steal ing _such _distress _sure ly _i _want _of _done _something _to _help _you $ |
| LPM | 1 | -72.55 | _oh _if _i _had _imagined _him _still _in _such _distress _sure ly _i _would _have _done _something _to _help _you $ |
| | 2 | -71.35 | _oh _if _i _had _imagined _him _still _in _such _distress _sure ly _i _might _have _done _something _to _help _you $ |
| | 3 | -69.29 | _oh _if _i _had _imagined _him _still _in _such _distress _sure ly _i _would _have _done _something _to _help _him $ |
| | 4 | -85.62 | _oh _if _i _had _imagined _him _still _in _such _distress _sure ly _i _won't _have _done _something _to _help _you $ |

We also notice in Table 9 that the LM probabilities correlate well with linguistic plausibility for texts of similar lengths (e.g., hypotheses of the same utterance), but not for texts with large differences in length. Motivated by this observation, we propose to quantify linguistic plausibility of an ASR model by measuring the LM token perplexity $\exp(\sum_{\boldsymbol{y}} \log p_{\mathbf{y}}(\boldsymbol{y}) / \sum_{\boldsymbol{y}} len(\boldsymbol{y}))$ of the hypotheses on the development set obtained using ASR-only greedy decoding. Results are shown in Table 10. The ground truth text has the lowest perplexity on both sets as expected. We also observe that while all models are worse on dev-other than on dev-clean, LPM exhibits the smallest perplexity difference between the two sets, demonstrating how it successfully distills knowledge from the LM.

Table 10: LM token perplexity of ground truth texts and hypotheses obtained with greedy decoding. A lower perplexity indicates more linguistically plausible decoding results.

| | $\mathcal{D}_l$ | $\mathcal{D}_u^s$ | LM perplexity | |
| --- | --- | --- | --- | --- |
| | | | dev-clean | dev-other |
| Ground Truth | N/A | N/A | 39.94 | 43.26 |
| Supervised | 100hr | N/A | 96.13 | 313.38 |
| Supervised | 460hr | N/A | 58.76 | 164.77 |
| Pseudo Label (ASR greedy) | 100hr | 360hr | 87.36 | 273.14 |
| Pseudo Label (ASR+LM stable) | 100hr | 360hr | 64.07 | 170.72 |
| Local Prior Matching | 100hr | 360hr | 61.73 | 159.72 |
| Local Prior Matching | 100hr | 860hr | 59.84 | 125.42 |

## 6 RELATED WORK

Our work builds on a large body of work in semi-supervised learning for ASR. Research in this direction can be classified based on the required modules and objectives used with unpaired data.

In Drexler & Glass (2018) and Karita et al. (2018), bi-encoder network architectures are used, which map text and speech to representations in a shared space with their corresponding encoder, and then apply a shared decoder to map from the shared space to the text space. Another line of work adds a TTS model (Tjandra et al., 2017; 2019; Baskar et al., 2019) or a text-to-encoding (TTE) model (Hayashi et al., 2018; Hori et al., 2019) in the loop of ASR training, which can be utilized for back-translation style data augmentation (Sennrich et al., 2016) or cycle-consistency training (Zhu et al., 2017). Liu et al. (2019) treats ASR as a generative model that conditions on speech instead of random noise vectors, and adopts the generative adversarial network (GAN) (Goodfellow et al., 2014) framework in order to improve the fidelity of ASR-generated texts (Liu et al., 2019). All the aforementioned methods involve additional modules that must be jointly optimized with the ASR model and require finding a careful balance between multiple training objectives. Moreover, some methods require gradient to flow through samples drawn from the ASR outputs, and hence rely on techniques like Gumbel-softmax (Jang et al., 2017), straight-through estimator (Bengio et al., 2013), or REINFORCE (Williams, 1992). The LPM objective can evaluate output quality yet is fully differentiable.

Knowledge distillation is also easy to used for semi-supervised learning (Cui et al., 2017; Li et al., 2019; Parthasarathi & Strom, 2019) because the teacher model can provide a training target for the unlabeled speech. Rather than learning from a teacher ASR model, LPM learns from a teacher LM.

Pseudo labelling (Veselỳ et al., 2017; Manohar et al., 2018; Kahn et al., 2019) is most similar to our approach. Pseudo labelling uses a seed ASR model to transcribe unlabeled speech. The ASR model is then trained on the predicted labels along with the original labeled data used to train the seed model. In addition to being well motivated theoretically, we highlight the key methodological advantages of LPM below. (1) While previous work uses fixed pseudo transcripts throughout training and focuses on data filtering, we demonstrate the benefit of generating hypotheses on-the-fly as the proposal model improves. This is efficient since we only use an end-to-end ASR model for beam search decoding. (2) We motivate and show empirically the benefit from considering multiple hypotheses rather than just the top hypothesis (Veselỳ et al., 2017; Kahn et al., 2019). (3) Manohar et al. (2018) consider multiple hypotheses in a numerator lattice and maximize the aggregated probability of the lattice regardless as to how the probability is allocated amongst the word sequences. In contrast, we verify in the LM ablation study that considering the allocation of the posterior probability amongst the hypotheses is important.

Aside from semi-supervised learning, this work is also related to unsupervised domain adaptation for ASR, where unlabeled speech of the target domain is provided. Unlike the proposed method, previous studies mainly focus on learning domain invariant features (Sun et al., 2017; Hsu & Glass, 2018; Meng et al., 2017; 2018; 2019) or data augmentation through learned speech transformation (Hsu et al., 2017; 2018).

## 7 CONCLUSION AND FUTURE WORK

We introduce local prior matching, a self-supervised learning objective for speech recognition, and demonstrate note-able reductions in WER when with the addition of unpaired audio and text. We also perform an extensive empirical study to demonstrate the importance of various configurations of LPM. While LPM is motivated by how humans learn to recognize speech, the proposed method can be applied to other sequence transduction tasks including machine translation (Sennrich et al., 2016) and text summarization (Nallapati et al., 2016), provided a good prior for the domain.

We consider two promising directions for future in self-supervised speech recognition with LPM and in general: (1) Empower the model with more capabilities, such as *learning* to select whether or not to use a sample for self-supervision. (2) Using more context signals for prior estimation which can consider previous sentences as well as inputs from other modalities.

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

## A  MORE COMPREHENSIVE COMPARISON WITH THE LITERATURE

Table 11: A more comprehensive comparison with semi-supervised ASR studies using LibriSpeech, including the performances of the baseline/topline supervised model used in each study, since they differ significantly across different papers. $\mathcal{D}_l$ and $\mathcal{D}_u$ denote the amount of paired and unpaired data used in each experiment, and (S)/(T)/(S+T) denote the type of the unpaired data, corresponding to speech/text/both, respectively. Experiments with the asterisk sign (*) contain results that are not reported in the original paper, but are obtained from the authors of the paper.

| | | $\mathcal{D}_l$ | $\mathcal{D}_u$ | LM | dev WER (%) clean | other | test WER (%) clean | other |
|---|---|---|---|---|---|---|---|---|
| (Hayashi et al., 2018) | Supervised | 100hr | N/A | None | 24.9 | - | 25.2 | - |
| | Supervised | 460hr | N/A | None | 11.4 | - | 11.8 | - |
| | BT | 100hr | 360hr (T) | None | 23.5 | - | 23.6 | - |
| | Supervised | 100hr | N/A | RNN-LM | 23.0 | - | 22.9 | - |
| | BT | 100hr | 360hr (T) | RNN-LM | 21.6 | - | 22.0 | - |
| (Liu et al., 2019) | Supervised | 100hr | N/A | None | 21.6 | - | 21.7 | - |
| | Crit-LM | 100hr | 360hr (T) | None | 19.1 | - | 19.2 | - |
| | Crit-LM | 100hr | 860hr (T) | None | 18.5 | - | 18.7 | - |
| | Supervised | 100hr | N/A | RNN-LM | 20.0 | - | 20.3 | - |
| | Crit-LM | 100hr | 360hr (T) | RNN-LM | 17.1 | - | 17.3 | - |
| | Crit-LM | 100hr | 860hr (T) | RNN-LM | 15.3 | - | 15.8 | - |
| (Hori et al., 2019) | Supervised | 100hr | N/A | None | 24.9 | - | 25.2 | - |
| | Supervised | 460hr | N/A | None | 11.4 | - | 11.8 | - |
| | Cycle-TTE | 100hr | 360hr (S) | None | 21.5 | - | 21.5 | - |
| | Supervised | 100hr | N/A | RNN-LM | 22.6 | - | 22.9 | - |
| | Cycle-TTE | 100hr | 360hr (S) | RNN-LM | 19.6 | - | 19.5 | - |
| (Baskar et al., 2019) | Supervised | 100hr | N/A | None | - | - | 21.0 | - |
| | Cycle-TTS | 100hr | 360hr (S) | None | - | - | 17.9 | - |
| | Cycle-TTS | 100hr | 360hr (S+T) | None | - | - | 17.5 | - |
| | Cycle-TTS | 100hr | 360hr (T) | RNN-LM | - | - | 17.0 | - |
| | Cycle-TTS | 100hr | 360hr (S) | RNN-LM | - | - | 16.8 | - |
| | Cycle-TTS | 100hr | 360hr (S+T) | RNN-LM | - | - | 16.6 | - |
| (Kahn et al., 2019) | Supervised | 100hr | N/A | None | 14.00 | 37.02 | 14.85 | 39.95 |
| | Supervised | 460hr | N/A | None | 7.20 | 25.32 | 7.99 | 26.59 |
| | PL (ASR)* | 100hr | 360hr (S) | None | 12.27 | 33.42 | 12.57 | 35.36 |
| | PL (ASR+LM) | 100hr | 360hr (S) | None | 9.30 | 28.79 | 9.84 | 30.15 |
| | PL (ASR+LM)* | 100hr | 860hr (S) | None | 9.03 | 26.03 | 9.44 | 27.25 |
| | PL (Ensemble) | 100hr | 360hr (S) | None | 8.60 | 27.78 | 9.21 | 29.29 |
| | Supervised | 100hr | N/A | ConvLM | 7.78 | 28.15 | 8.06 | 30.44 |
| | Supervised | 460hr | N/A | ConvLM | 3.98 | 17.00 | 4.23 | 17.36 |
| | PL (ASR)* | 100hr | 360hr (S) | ConvLM | 6.19 | 23.53 | 6.81 | 24.99 |
| | PL (ASR+LM) | 100hr | 360hr (S) | ConvLM | 5.73 | 22.54 | 6.35 | 24.13 |
| | PL (ASR+LM)* | 100hr | 860hr (S) | ConvLM | 6.31 | 21.87 | 6.84 | 23.29 |
| | PL (Ensemble) | 100hr | 360hr (S) | ConvLM | 5.37 | 22.13 | 5.93 | 24.07 |
| This work | Supervised | 100hr | N/A | None | 14.00 | 37.02 | 14.85 | 39.95 |
| | Supervised | 460hr | N/A | None | 7.20 | 25.32 | 7.99 | 26.59 |
| | LPM | 100hr | 360hr | None | 8.85 | 26.33 | 9.21 | 27.74 |
| | LPM | 100hr | 860hr | None | 8.08 | 21.52 | 8.37 | 22.89 |
| | Supervised | 100hr | N/A | ConvLM | 7.78 | 28.15 | 8.06 | 30.44 |
| | Supervised | 460hr | N/A | ConvLM | 3.98 | 17.00 | 4.23 | 17.36 |
| | LPM | 100hr | 360hr | ConvLM | 5.69 | 20.22 | 5.99 | 20.93 |
| | LPM | 100hr | 860hr | ConvLM | 5.39 | 14.89 | 5.78 | 16.27 |

## B  ADDITIONAL RESULTS

Table 12: Results of varying LPM weight $\alpha$.

| $\alpha$ | dev WER (%) | |
|---|---|---|
| | clean | other |
| 2e-2 | 10.86 | 31.59 |
| 5e-2 | 10.08 | 28.92 |
| 1e-1 | 9.24 | 27.62 |
| 2e-1 | 9.00 | 26.47 |
| 5e-1 | 9.41 | 26.56 |

## C  CHARACTER ERROR RATES

Table 13: Results of baselines and the proposed methods.

| | $\mathcal{D}_l$ | $\mathcal{D}_u^s$ | LM | dev CER (%) | | test CER (%) | |
|---|---|---|---|---|---|---|---|
| | | | | clean | other | clean | other |
| Supervised | 100hr | N/A | None | 6.20 | 20.27 | 6.80 | 22.14 |
| Supervised | 460hr | N/A | None | 2.86 | 13.06 | 3.37 | 13.73 |
| Local Prior Matching | 100hr | 360hr | None | 3.79 | 14.00 | 3.87 | 14.81 |
| Local Prior Matching | 100hr | 860hr | None | 3.52 | 11.14 | 3.60 | 12.08 |
| Supervised | 100hr | N/A | ConvLM | 3.83 | 17.03 | 3.86 | 18.52 |
| Supervised | 460hr | N/A | ConvLM | 1.65 | 9.51 | 1.79 | 9.47 |
| Local Prior Matching | 100hr | 360hr | ConvLM | 2.65 | 11.56 | 2.81 | 11.96 |
| Local Prior Matching | 100hr | 860hr | ConvLM | 2.51 | 8.70 | 2.70 | 9.70 |

