# OpenReview forum: "Self-Supervised Speech Recognition via Local Prior Matching"
_ICLR.cc/2020/Conference — Reject_

### Official Review · AnonReviewer2 · 2019-10-22
**Official Blind Review #2**

**Rating:** 6

**Review:**

Overview:
This paper is dedicated to proposing a self-supervised objective, local prior matching (LMP), for speech recognition. This approach can take advantage of vase quantities of unlabeled speech data. What' more, the objective is simple to implement and theoretically well-motivated. In the paper, based on a supervised pretrained model, it then finetunes 360 hours with unlabeled data and LPM reduces the WER consistently. They also conduct extensive ablation experiments to show the effect of their self-supervised approach.

Strength Bullets:
1. I think this self-supervised learning objective (LMP) is very novel. The motivation that the source of indirect supervision on processing unlabeled speech comes from prior knowledge about the world and the context of the speech makes sense to me. The author combines the Bayesian method to build the model which is aligned with the motivation. They also provide clear and well-organized derivations.
2. The paper performs extensive ablation studies over all components, including beam size, mixing ratio, LPM weights, model update strategies, model initialization, length filtering and choice of language models. It provides convincing evidence of the effect of each component.
3. It provides interesting experiments results to study the relationship between the amount of unlabelled data for self-supervision and final performance. And LPM can surpass the performance of using 360 hours of labeled data by taking advantage of about twice the amount of unlabeled data.


Weakness Bullets:
1. The paper only evaluates their method on LibriSpeech dataset. Although this dataset is popular, one or two more datasets will be more convincing.
2. For bayesian based methods, it is well known that it performs badly in high dimensional space. The reason is that we can not sample enough data points to obtain a good posterior estimation. Could you provide more analysis about the quality of posterior with different amount of sampled data?
3. For the experiment between the amount of unlabelled data for self-supervision and final performance, it would be better the author can provide a curve with more results.

Recommendation:
I think it is a good paper. The proposed approach is useful. This is a weak accept.

**Experience Assessment:**

I have read many papers in this area.

**Review Assessment: Checking Correctness Of Derivations And Theory:**

I assessed the sensibility of the derivations and theory.

**Review Assessment: Checking Correctness Of Experiments:**

I assessed the sensibility of the experiments.

**Review Assessment: Thoroughness In Paper Reading:**

I read the paper at least twice and used my best judgement in assessing the paper.

---

> ### Author Response · Authors · 2019-11-07
> **Response to AnonReviewer2**
>
> We thank the reviewer for the thoughtful comments. Below are our itemized responses to address the concerns.
>
> Q1: The paper only evaluates their method on LibriSpeech dataset. Although this dataset is popular, one or two more datasets will be more convincing.
>
> A1: We completely agree with the reviewer more datasets would certainly support our claims. However, we are limited by publicly available ASR datasets which are sufficiently large and challenging to demonstrate semi-supervised learning at scale, which is a condition researchers have found difficult to achieve any improvement [1].
>
> We point out that prior work in this domain has used LibriSpeech as a sole benchmark [2; 3; 4; 5]. Also, the improvement from running our algorithm on LibriSpeech is more than 82% WER recovery rate, which is notably larger than prior work. Finally, we plan to test Local Prior Matching on other datasets and domains in future work.
>
> [1] Drexler, Jennifer, and James Glass. "Combining end-to-end and adversarial training for low-resource speech recognition." 2018 IEEE Spoken Language Technology Workshop (SLT). IEEE, 2018.
> [2] Kahn, Jacob, Ann Lee, and Awni Hannun. "Self-Training for End-to-End Speech Recognition." arXiv preprint arXiv:1909.09116 (2019).
> [3] Hori, Takaaki, et al. "Cycle-consistency training for end-to-end speech recognition." ICASSP 2019-2019 IEEE International Conference on Acoustics, Speech and Signal Processing (ICASSP). IEEE, 2019.
> [4] Liu, Alexander H., Hung-yi Lee, and Lin-shan Lee. "Adversarial training of end-to-end speech recognition using a criticizing language model." ICASSP 2019-2019 IEEE International Conference on Acoustics, Speech and Signal Processing (ICASSP). IEEE, 2019.
> [5] Hayashi, Tomoki, et al. "Back-translation-style data augmentation for end-to-end ASR." 2018 IEEE Spoken Language Technology Workshop (SLT). IEEE, 2018.
>
>
> Q2: For bayesian based methods, it is well known that it performs badly in high dimensional space. The reason is that we can not sample enough data points to obtain a good posterior estimation. Could you provide more analysis about the quality of posterior with different amount of sampled data?
>
> A2: This is a great question. Unfortunately, since we are working with a real dataset, we do not have access to the ground truth posterior p(y|x) and therefore cannot directly evaluate the quality of our proposed estimator. However, Table 1 does provide indirect analysis about the quality of the estimator with respect to different amounts of sampled data, with the assumption that the performance should be better if the ASR model distribution is matched to a better posterior estimator. As discussed in Section 4.1, the estimator usually gets better with more samples.
>
> We hypothesize that the reason this estimator works well with few samples (from 1 to 16) in this high dimensional text sequence space is because the posterior p(y|x) is extremely spiky with only a few y having non-negligible probability mass. This assumption is commonly acknowledged and enables estimation of the partition function p(x) = \int_y p(x, y) using beam search hypotheses in many ASR studies [3;4;5].
>
> [3] Collobert, R., Hannun, A. & Synnaeve, G.. (2019). A fully differentiable beam search decoder. Proceedings of the 36th International Conference on Machine Learning, 2019.
> [4] Povey, Daniel, et al. "Boosted MMI for model and feature-space discriminative training." ICASSP, 2008.
> [5] Povey, Daniel, and Philip C. Woodland. "Minimum phone error and I-smoothing for improved discriminative training." ICASSP, 2002.
>
>
> Q3:  For the experiment between the amount of unlabelled data for self-supervision and final performance, it would be better the author can provide a curve with more results.
>
> A3: We thank the reviewer for the great suggestion, and we are running more experiments varying the amount of unlabeled data at this moment. We will report the numbers once the experiments are finished.

---

> > ### Author Response · Authors · 2019-11-15
> > **Updated response for Q3 (experiments varying the amount of unlabeled speech)**
> >
> > We created another three subsets: (1) train-clean-180, (2) train-other-180, and (3) train-other-360 to study the LPM performances with different unlabeled speech sizes. In particular, (1) contains 180 hrs of speech sampled from train-clean-360, (3) contains 360 hrs of speech sampled from train-other-500, and (2) contains 180hrs sampled from (3). To sample a subset, we randomly chose a subset of speakers and include all their utterances in it.
> >
> > We ran experiments with five different sizes of unlabeled speech (180/360/540/720/860 hrs), following the hyperparameters described in Section 3, with a mixing rate of 1:4, a beam size of 4, and a LPM weight of 0.2. For each size, we train for 1.6M steps and report the average WER over three runs below.
> >
> > Unlabeled Speech Size      		|  dev-clean WER	|  dev-other WER
> > Baseline (0 hrs)		                |  14.00%                |  37.02%
> > clean 180 hrs 			        |  10.41%		 |  29.55%
> > clean 360 hrs 			        |  9.00%		         |  26.69%
> > clean 360 hrs + other 180 hrs	|  8.74%		         |  23.22%
> > clean 360 hrs + other 360 hrs	|  8.74%		         |  22.66%
> > clean 360 hrs + other 500 hrs	|  8.77%		         |  22.31%
> >
> > The results demonstrate a favorable trend, where consistent improvements can be observed on dev-other, For dev-clean, the improvement seems to saturate after using 540hr of unlabeled speech. However, we want to emphasize that this could have resulted from a sub-optimal learning rate schedule for larger datasets, as bigger datasets can benefit from more training steps and slower learning rate decay. This is evidently shown in Section 4.5, where the final result with 860 hrs of speech reported in Table 7 (trained for 3.2M steps) are much better than the result here (8.08% vs 8.77%).

---

### Official Review · AnonReviewer1 · 2019-10-23
**Official Blind Review #1**

**Rating:** 3

**Review:**

This work proposed a distillation approach which use ASRs to generate hypotheses for unsupervised data, run a LM to get probability for the hypothesis, and perform distillation with the resulting probability. The ASRs being used for generating hypotheses can be either a model trained with the supervised data or the student model, and can switch between the two during training. In the experiments, ASR models are pre-trained with the subset of Librispeech data and use the rest of Librispeech data as unsupervised data, and the LM is trained with Librispeech LM data. The experiments shown the proposed approach improve baseline model trained with the Librispeech subset significantly.

The use of LM to provide soft target is a good idea as LMs can utilize unsupervised text data as opposed to the requirement of training a strong teacher model with paired data, and can be easily integrated with existing distillation approaches for ASRs. The switching to the student model for generating hypotheses when it outperforms the pre-trained ASR also makes a good sense. The overall novelty however is a bit limited compared to the existing work, as the major contribution is to propose to use LMs as teacher rather than ASRs, with the rest of the design to be similar to existing works.

The paper relates their method to self-supervised learning, yet I find it having stronger correlation with existing distillation approaches, and can be better understood through the distillation perspective.

**Experience Assessment:**

I have published in this field for several years.

**Review Assessment: Checking Correctness Of Derivations And Theory:**

I assessed the sensibility of the derivations and theory.

**Review Assessment: Checking Correctness Of Experiments:**

I carefully checked the experiments.

**Review Assessment: Thoroughness In Paper Reading:**

I read the paper at least twice and used my best judgement in assessing the paper.

---

> ### Author Response · Authors · 2019-11-07
> **Response to AnonReviewer1**
>
> We thank the reviewer for the thoughtful comments. Below are our itemized responses to address the concerns.
>
>
> Q1: The overall novelty however is a bit limited compared to the existing work, as the major contribution is to propose to use LMs as teacher rather than ASRs, with the rest of the design to be similar to existing works. The paper relates their method to self-supervised learning, yet I find it having stronger correlation with existing distillation approaches, and can be better understood through the distillation perspective.
>
> A1: Despite the similarity in the form of the objectives, the proposed LPM method and knowledge distillation are motivated very differently, and therefore differ a lot in their capability as well as theoretical soundness. The two methods are different for the following reasons:
>
> 1. We have found that our approach is consistently superior to weak knowledge distillation (Table 1). When the beam size is set to 1, our approach yields self-training [3; 4; 5], which is identical to weak distillation [1; 2] where the teacher distribution is replaced by its mode.
>
> 2. Unlike knowledge distillation, our approach is Bayesian and derives a tractable posterior estimator. Compared with the heuristic of weak knowledge distillation, we provide a theoretically well-motivated approach that is simple to implement, but also superior in performance.  Furthermore, our framework is flexible: the proposal model does not have to be tied to the ASR model, and we can also make the estimator more accurate by adding a TTS component to re-weigh acoustic plausibility.
>
> We will incorporate the discussion into the paper if the reviewers find it helpful to distinguish knowledge distillation from our proposed approach.
>
> [1] Kim, Yoon, and Alexander M. Rush. "Sequence-level knowledge distillation." arXiv preprint arXiv:1606.07947 (2016).
> [2] Li, Bo, et al. "Semi-supervised training for End-to-End models via weak distillation." ICASSP (2019).
> [3] Veselý, Karel, Lukás Burget, and Jan Cernocký. "Semi-Supervised DNN Training with Word Selection for ASR." INTERSPEECH (2017).
> [4] Manohar, Vimal, et al. "Semi-supervised training of acoustic models using lattice-free MMI." ICASSP (2018).
> [5] Kahn, Jacob, Ann Lee, and Awni Hannun. "Self-Training for End-to-End Speech Recognition." arXiv preprint arXiv:1909.09116 (2019).

---

### Official Review · AnonReviewer3 · 2019-10-30
**Official Blind Review #3**

**Rating:** 3

**Review:**

This paper propose local prior matching to leverage a language model to use unlabeled speech data to improve an ASR system. This is a worthy goal. The details of the proposal were a bit hard for me to understand. The proposed method reminded me of "posterior regularization" (K. Ganchev et al. 2010), but I could not understand Section 2.2 well enough to draw a direct link. I encourage the authors to condense 2.2 and make it clearer what, exactly, Local Prior Matching is.

The paper presents extensive, interesting results. I do want to point that they seem to be considerably off of the LibriSpeech state of the art, e.g. see K. Irie et al. Interspeech 2019.

**Experience Assessment:**

I have read many papers in this area.

**Review Assessment: Checking Correctness Of Derivations And Theory:**

I assessed the sensibility of the derivations and theory.

**Review Assessment: Checking Correctness Of Experiments:**

I assessed the sensibility of the experiments.

**Review Assessment: Thoroughness In Paper Reading:**

I read the paper at least twice and used my best judgement in assessing the paper.

---

> ### Author Response · Authors · 2019-11-07
> **Response to AnonReviewer3**
>
> We thank the reviewer for the thoughtful comments. Below are our itemized responses to address the concerns.
>
>
> Q1: The details of the proposal were a bit hard for me to understand. The proposed method reminded me of "posterior regularization" (K. Ganchev et al. 2010), but I could not understand Section 2.2 well enough to draw a direct link. I encourage the authors to condense 2.2 and make it clearer what, exactly, Local Prior Matching is.
>
> A1: We thank the reviewer for the suggestion on writing. We will clarify Section 2 of the paper. If the reviewer has specific parts in mind that are not clear, we will gladly address them.
>
> In a nutshell, we propose a training objective for unlabeled speech. Our approach is like pseudo-labeling, but instead of using the top-1, we use top-k with beam search, and re-weight this top-k by the LM score. We call it the “local prior” because it gives a distribution proportional to the prior (LM), but only in a region close to the ground truth, where p(speech | text) is high. We then train an ASR system using unlabeled speech by matching the ASR model distribution with this target distribution, and hence the proposed training objective is termed “local prior matching.”
>
> Both posterior regularization (PR) [1] and our work aim to incorporate implicit supervision, but the methods differ significantly. PR focuses on incorporating domain knowledge (e.g., in POS tagging there must be at least one noun and one verb in the output) through adding *handcrafted* and *linear* constraints to the posterior distribution family. Optimization of PR is done with an EM algorithm.
>
> On the other hand, we propose a Bayesian-based method, where the implicit supervision from the language model corresponds to the prior in the Bayesian framework. Unlike PR, there is no limitation on what models can be used for parameterizing the prior distribution. Hence, we can use a very strong prior model that incorporates all the prior knowledge (e.g., in the POS tagging example, any sequence with no verb and no noun should have extremely low prior probability). One of our main contributions is proposing a tractable and theoretically justified posterior estimator utilizing a strong prior distribution model for sequence transduction tasks.
>
> [1] Ganchev, Kuzman, Jennifer Gillenwater, and Ben Taskar. "Posterior regularization for structured latent variable models." Journal of Machine Learning Research 11.Jul (2010): 2001-2049.
>
>
> Q2: The paper presents extensive, interesting results. I do want to point that they seem to be considerably off of the LibriSpeech state of the art, e.g. see K. Irie et al. Interspeech 2019.
>
> A2: We thank the reviewer for pointing out the reference and we are also aware of those work. As described in our paper, we base our model on [2] because it is light-weight and efficient compared to the RNN-based encoders used in [3; 4], while achieving comparable performances. Below we list our baseline model results and those from some very recent literature using seq2seq+attention ASR models trained on LibriSpeech train-clean-100. Our baseline model is on-par with the second place and not far from the best results.
>
> *Baseline Performances (WER)*
> Paper  |  test-clean |  test-other
> Ours   |  14.85%     |  39.95%
> [5]       |  25.2%       |  (not reported)
> [6]       |  21.0%       |  (not reported)
> [3]       |  14.7%       |  40.8%
> [4]       |  12.9%       |  35.5%
>
> We also point out that the reference the reviewer mentioned [4] was concurrent work published just one week before the ICLR submission deadline. Since the focus of our paper is semi-supervised learning and not on achieving the best possible baseline, we feel that the important results are the amount of improvement from the baseline and the gap reduced from using a larger labeled dataset (WER recovery rate, WRR). In that respect, our proposed method demonstrates superior performance compared to the literature as shown below (complete results are in Table 11), while being extremely simple to implement and theoretically well-justified.
>
> *Proposed Method Performances with 360hr of unlabeled speech*
> WRR = (WER(sup. 100) - WER(proposed)) / (WER(sup. 100) - WER(sup. 460))
> Paper |  test-clean WER  |  test-clean WRR
> Ours  |  9.21%           |  82.22%
> [5]      |  21.5%           |  27.6%
> [6]      |  17.5%           |  38.0%
>
>
> [2] Hannun, Awni, et al. "Sequence-to-Sequence Speech Recognition with Time-Depth Separable Convolutions." Interspeech (2019).
> [3] Lüscher, Christoph, et al. "RWTH ASR systems for LibriSpeech: Hybrid vs Attention." Interspeech (2019).
> [4] Irie, Kazuki, et al. "On the Choice of Modeling Unit for Sequence-to-Sequence Speech Recognition." Interspeech (2019).
> [5] Hori, Takaaki, et al. "Cycle-consistency training for end-to-end speech recognition." ICASSP (2019).
> [6] Baskar, Murali Karthick, et al. "Self-supervised Sequence-to-sequence ASR using Unpaired Speech and Text." Interspeech (2019).

---

### Decision · Program_Chairs · 2019-12-19

**Decision:**

Reject

**Comment:**

The paper proposed local prior matching that utilizes a language model to rescore the hypotheses generate by a teacher model on unlabeled data, which are then used to training the student model for improvement. The experimental results on Librispeech is thorough. But two concerns on this paper are: 1) limited novelty: LM trained on large tex data is already used in weak distillation and the only difference is the use of multiply hypotheses. As pointed out by the reviewers, the method is better understood through distillation even though the authors try to derive it from Bayesian perspective. 2) Librispeech is a medium sized dataset, justifications on much larger dataset for ASR would make it more convincing.